# Effects of Mining on Urban Environmental Change: A Case Study of Panzhihua

Xiaoai Dai [1] , Wenyu Li [1], Zhilong Liu [1], Chenbo Tong [1], Cheng Li [1], Jianwen Zeng [1], Yakang Ye [2,*], Weile Li [1] , Yunfeng Shan [1] , Jiayun Zhou [2], Junjun Zhang [2], Li Xu [2], Xiaoli Jiang [2], Huihua Ruan [3], Jinbiao Zhang [3] and Wei Huang [3]

1    College of Earth Science, Chengdu University of Technology, Chengdu 610059, China
2    Institute of Multipurpose Utilizationg of Mineral Resources, China Academy of Geological Science, Chengdu 610042, China
3    Guangdong Meteorological Observation Data Center, Guangzhou 510080, China
*    Correspondence: yyakang@mail.cgs.gov.cn

**Abstract:** The environment supplies water, land, biological resources, and climate resources for people's daily life and development, dramatically affecting the subsistence and development of human beings. Panzhihua City is a representative resource-based industrial city of southwestern China. The abundant mineral resources provide the material basis for the city's development. However, while the overdevelopment of the past decades has provided the preconditions for its rapid economic growth, it has also inevitably had a huge impact on its environmental quality and land use structure. In this study, the landsat remote sensing images, terrain data, socio-economic data, and mining resources exploitation data of Panzhihua were used to extract the NDVI (Normalized Difference Vegetation Index), NDBSI (Normalized Difference Build and Soil Index) and LST (Land Surface Temperature) of the past 20 years at 5-year intervals. We normalized four indicators by Principal Component Analysis to derive a remote sensing ecological index of each factor and build the Remote Sensing-based Ecological Index evaluation model. This research quantified the changes in environmental quality in the past 20 years through the range method, showing that the environmental quality of Panzhihua City first declined and then increased slowly. This research also analyzed the influence of land use types, terrain, mining area, and socio-economy on the environmental quality of Panzhihua City by grey relational analysis and buffer analysis. It is found that with the influence of its unique topographical factors and economic aspects, the environmental quality of Panzhihua City changed to varying degrees. The results provide a reliable basis for the future environmental planning of Panzhihua City and a reference for the ecological restoration of mining areas with different mineral species accurately.

**Keywords:** RSEI; grey relational analysis; environmental quality; Panzhihua

## 1. Introduction

The environment includes the ensemble of human activities, biomass, geological resources, and the composition of the basic elements of life, water, and air in a certain area. Environmental quality has an extremely important impact on the development of human social activities [1]. In 2013, the United Nations Environment Program affirmed and shared the experience of different countries in environmental governance, and set goals for international environmental quality development. The 2016 United Nations International Conference affirmed and supported China's remarkable achievements in environmental conservation in recent years, and shared with other countries the wisdom and methods presented by China in this regard. With the coordinated development of the economy and environment, environmental quality monitoring has become the basis for scientifically planning the future development of a city. Using the scientific method to assess long-term environmental quality changes has become a top research topic [2].

Since the 1960s, scholars have begun to research changes in regional environmental quality [3]. With the development of Earth observations from remote sensing satellite and their wide application in various fields, the evaluation of the quality of the environment in a region, including province, country, and project, has become an important method for protecting it [4]. In 2006, the China Environmental Monitoring Centre issued the Technical Specifications for the Evaluation of the Environment, which is regarded as an industry norm and standard for monitoring and analysis of forests, grasslands, cities, and water [5]. But the evaluation indicators used in this method are not suitable for long-term regional evaluation and monitoring. The 3S (RS (Remote sensing), GIS (Geography information systems), and GPS (Global positioning systems)) technology can used to extract remote sensing images and create a comprehensive index evaluation method to analyze and monitor environmental quality in the districts and counties [6]. A model combining grey relational analysis and environmental quality was built [7]. The analytic hierarchy process was used to analyse the changes in the quality of the environment and the relevant factors affecting the ecology of a river basin [8]. Xu (2013) evaluated and monitored the quality of the regional environment using a long-term series of the remote sensing ecological index (RSEI) based on greenness (NDVI), humidity (WET), dryness (NDBSI), and surface temperature (LST) calculated from remote sensing image bands [9]. These indices are closely related to daily human life and can objectively, quickly and quantitatively evaluate and monitor the quality of the regional environment. This method has been effectively applied in different regions [10]. With the rapid development of geography and remote sensing science, the research on the quality of the environment in different regions has been improved with the help of these disciplines [11]. Evaluation methods developed from single factors have been extended to multiple factors at different scales, such as in the case of the ecological remote sensing index, which has promoted in-depth research in this area and provided local governments with a scientific reference for ecological planning and governance [12].

Combining the impact of land use and other factors on the quality of the environment has become a new area of research [13]. Scholars successively adopted various methods, combining the natural environment and ecological impact factors by using RS and GIS methods. The suitability evaluation model of the interaction between land use and urban environmental quality was constructed [14]. A detailed analysis of the driving forces of ecosystem change, considering future development, provides a relevant reference value for the development of the regional environment [15]. Among them, the gray system refers to the information factors, factor relationships, and correlation systems that are not completely determined [16]. The factor analysis method, Grey Relational Analysis (GRA), is used to evaluate if two or more factors of the system have the same change in the process of development. In this case, the degree of correlation between the factors is considered to be large. Compared with correlation analysis and regressive analysis, the Grey Relational Analysis model provides a quantitative method suitable for dynamic analysis. The GRA model has been widely used in various fields such as environmental protection and agricultural development [17].

The unreasonable use of resources by the ecosystem of resource-based cities leads to excessive depletion of the environment and the deterioration of the quality of the natural ecological environment. In the research on ecosystem services of resource-based cities, scholars explored the carrying capacity of mining cities from the perspective of the relationship between environment and resources [18,19], or analyze the spatio-temporal variation through the remote sensing index [20]. Panzhihua is rich in minerals. Vanadium and titanium magnetite in the region has large reserves, reliable resources, concentrated distribution, complete varieties, comprehensive utilization value and other characteristics. However, most of the mines are open-pit mining, with a large scale of mining and a large number of mining sites. The current situation of mineral resources development is generally good, the 2014 remote sensing survey found that the coal mining order is good, mining irregularities and mining scale of small private coal mines have been shut down. At present,



only state-owned coal mines with good mining order and private coal mines with large mining scale are still mining; iron ore mines still have suspected illegal mining activities. Repeated monitoring of the vast majority of mine development order in the area continues to be good, but the illegal development of iron ore, copper ore, phosphate ore has an expanding trend. The area is rich in tailings resources, but the degree of comprehensive recycling is too low, and has considerable potential for development [21]. Many scholars pay attention to the ecological vulnerability and ecological carrying capacity of Panzhihua City [22], but few studies have analyzed the causes of the changes in the multi-mining areas by means of spatial analysis.

In this study, we combined remote sensing data and mine survey data from 2000 to 2020. The remote sensing ecological index was used as an evaluation method to establish a long temporal series of environmental quality evaluation systems for the research area of Panzhihua City, Sichuan Province, China. The research results can provide relevant reference values for the construction of an ecological civilization and for decision-making for the future urban planning and delineation of ecological function zones in Panzhihua City.

## 2. Materials and Methods

### 2.1. Study Area

Panzhihua City is located in southwest Sichuan, China, bordering Yunnan Province across the Jinsha River (Figure 1). Its geographical coordinates are between $101°08'{\sim}102°15'$N and $26°05'{\sim}27°21'$E and the entire urban administrative area is about 7401 square kilometers. It has abundant mineral resources, including vanadium, titanium, and so on [23]. Among them, the associated vanadium, titanium, graphite, and other resources are at the forefront of China [24]. Vanadium and titanium magnetite is one of the three famous polymetallic deposits in China, with vanadium, titanium, scandium, gallium, cobalt, nickel, and other valuable elements. Panzhihua City has high mountains, deep valleys, a staggered basin, and large altitudinal differences. It is high in the north and low in the south, with an elevation range of 954 m~4143 m [25]. Due to its temperate continental climate, the rivers and valleys in the region are intertwined, and the Yalong River and Jinsha River intersect here. Panzhihua City is the first city in the upper reaches of the Yangtze River.

### 2.2. Data

#### 2.2.1. Remote Sensing Data

The remote sensing ecological index was calculated from the thematic mapper (TM) and multi-spectral scanner (MSS) of Landsat 5 and operational land imager (OLI) of Landsat 8 satellites, which were jointly launched by the National Aeronautics and Space Administration and the United States Geological Survey. Panzhihua City's area can be covered by three scenes with orbit numbers of, in order, 130-041, 130-042, and 131-041. The sources of the data are detailed in Table 1.

We compared the key parameters (band, wavelength, spatial resolution) of the Landsat 5 and 8 satellites. In order to accurately extract the remote sensing ecological indices in the later stage and reduce unnecessary errors, we selected high-quality remote sensing images with fewer clouds and as close as possible in imaging times [26].

#### 2.2.2. Ancillary Data

The main ancillary data used in this paper were digital elevation model (DEM) data, mining point data, and statistical yearbook data.

(1) DEM data: we used the 30-m resolution data from the product "GDEMV3 30M resolution digital elevation data".
(2) Mining area distribution data: the mining data in this paper were from the mineral resource distribution, the latest mineral resource planning texts, and tables provided by Sichuan Geological Survey Institute.

(3)   Statistical yearbook data: the statistical yearbook data were downloaded from the Bureau of Statistics of Panzhihua City and mainly included average annual rainfall, total sunshine hours, arable land, the built-up area of Panzhihua City, and the gross value of primary, secondary and tertiary industries in Panzhihua City.

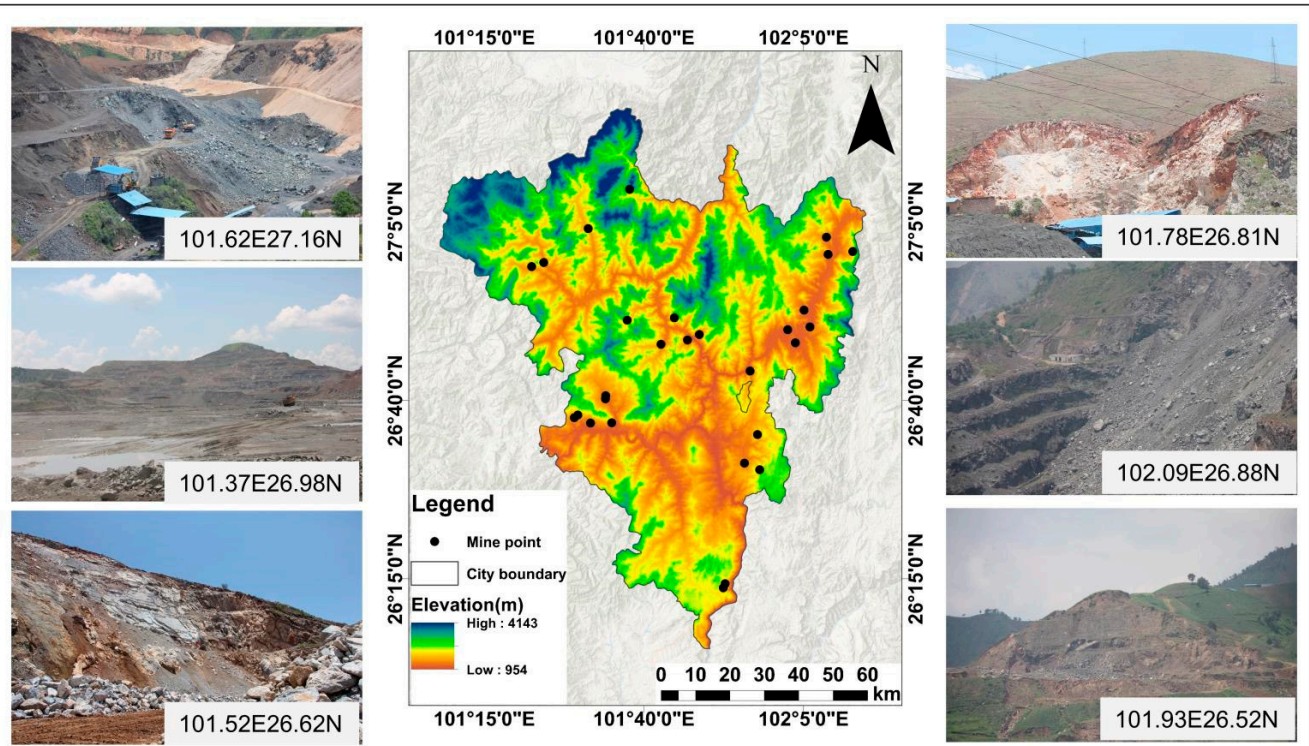

**Figure 1.** Location of the study area.

**Table 1.** Data information and data sources.

| Data | Spatial Resolution | Data Source |
|---|---|---|
| Remote sensing imagery data-Landsat 5/8 | 30 × 30 m | Geospatial Data Cloud (http://www.gscloud.cn/) (accessed on 20 January 2020.) |
| Land use data | 30 × 30 m | Google Earth Engine platform (accessed on 9 December 2021.) |
| Digital elevation model (DEM) data | 30 × 30 m | Geospatial Data Cloud (http://www.gscloud.cn/) (accessed on 20 January 2020.) |
| Mine environment monitoring data in Sichuan Province Database | Forms for Report | Sichuan Geological Survey Institute (accessed on 31 July 2019) |
| Statistical yearbook data | Forms for Report | the Bureau of Statistics of Panzhihua City (http://tjj.panzhihua.gov.cn/) (accessed on 28 January 2022.) |

All spatial data were unified under the WGS_1984_UTM_Zone_47N projection coordinate system, and the raster data were resampled to 30 m spatial resolution.

### 2.3. Principle and Construction of the Remote Sensing Ecological Index

The construction of this research method is completely based on remote sensing technology. The four important indicators of greenness (NDVI), wetness (WET), dryness (NDBSI), and surface temperature (LST), which are closely related to daily human activities,



were calculated from the remote sensing images [27]. The remote sensing ecological index RSEI can be calculated by using the following function expression:

$$RSEI = (Greenness, Wetness, \ Heat, Dryness)$$
$$RSEI = f(NDVI, WET, LST, NDSI) \tag{1}$$

### 2.3.1. Greenness Index

The NDVI is an important parameter indicating vegetation growth and density distribution and reflecting the close relationship between plant biomass, leaf area index and vegetation cover, and is used to monitor the changes in vegetation growth and the coverage in the study area, which is currently the most used indicator in vegetation condition indicators [28]. This paper selected NDVI as one of the indexes to construct the evaluation system of the remote sensing ecological index. The formula is as follows:

$$NDVI = \frac{\rho_{nir} - \rho_{red}}{\rho_{nir} + \rho_{red}} \tag{2}$$

where $\rho_{nir}$ represents the near-infrared band reflection value, $\rho_{red}$ represents the infrared band reflection value.

### 2.3.2. Humidity Index

The moisture index WET reflects the moisture information in the soil and vegetation. A low WET indicates that the land is degraded and the vegetation cover is reduced. A high WET means that the soil water content is high and the surface vegetation cover is abundant. In this paper, the humidity of the image was extracted through the tasseled cap transformation [29], calculated as:

$$WET = C_1 B_1 + C_2 B_2 + C_3 B_3 + C_4 B_4 + C_5 B_5 + C_6 B_6 \tag{3}$$

where $B_1$ represents the blue band, $B_2$ represents the green band, $B_3$ represents the red band, $B_4$ represents the near-infrared band, $B_5$ represents the mid-infrared band 1, $B_6$ represents the mid-infrared band 2; C1, C2, C3, C4, C5 and C6 represent sensor parameters. The parameters differ based on the image sensor [30]. In the case of the TM sensor from Landsat 5, C1 = 0.031, C2 = 0.2021, C3 = 0.3012, C4 = 0.1594, C5 = −0.6806, C6 = −0.6109. For the OLI sensor from Landsat 8, the parameter values are C1 = 0.1511, C2 = 0.1973, C3 = 0.3283, C4 = 0.3407, C5 = −0.7117, C6 = −0.4559.

### 2.3.3. Heat Index

For TM data, this research used radiometric calibration parameters to convert the image element grayscale value of thermal infrared (band 6) into the radiometric brightness value at the sensor, and then obtained the surface temperature (LST) after correcting by the land surface emissivity:

$$L_6 = gain \times DN + bias \tag{4}$$

$$Pv = \left[ \frac{NDVI \times NDVI_{soi1}}{NDVI_{veg} \times NDVI_{soi1}} \right] \tag{5}$$

$$\varepsilon = 0.004 P_V + 0.986 \tag{6}$$

$$T = \frac{K_2}{ln\left(\frac{K_1}{L_6} + 1\right)} \tag{7}$$

$$LST = T / [1 + (\lambda \times T / \rho)] \times ln\varepsilon \tag{8}$$

where DN represents the pixel gray value; gain = 0.055, bias = 1.18243; the calibration parameters are $K_1 = 607.76 \ w/(m^2 \cdot sr \cdot um)$ and $K_2 = 1260.56$ K; $\lambda = 11.45$ μm is the central wavelength of thermal infrared band of TM image; $\rho = 1.438 \times 10^{-2}$ K ; $\varepsilon$ is the surface

emissivity, whose value can be estimated by the NDVI threshold method proposed by Sobrino [31]. $p_v$ is the vegetation cover; NDVI is the normalized vegetation index taking $NDVI_{veg}$ as 0.7 and $NDVI_{soil}$ as 0.05; when the NDVI value of a pixel was greater than 0.7, the value was set to 1; when the NDVI value was less than 0.05, the value was set to 0.

For OLI sensor data, the method of obtaining surface temperature is usually inversed by atmospheric correction [32]. In order to eliminate the energy absorbed by the atmosphere in the process of solar radiation and to obtain a more accurate surface radiation temperature, the constant of Planck's formula and seek $T_s$ were used to erase this part of the effect.

$$WL_{10} = \left[\varepsilon B(T_s) + (1 - \varepsilon)L_\downarrow\right]\tau + L_\uparrow \tag{9}$$

$$B(T_S) = \left[L_\lambda - L_\uparrow - \tau(1 - \varepsilon)L_\downarrow\right]\frac{1}{\tau\varepsilon} \tag{10}$$

$$T_S = \frac{K_2}{ln\left(\frac{K_1}{B(T_S)} + 1\right)} \tag{11}$$

where $WL_{10}$ is the thermal infrared radiation brightness value received by the satellite sensor, $L_\uparrow$ represents the upward radiation brightness of the atmosphere, $L_\downarrow$ represents the energy reflected by the atmosphere after the downward radiation reaches the ground; $\varepsilon$ represents the surface emissivity; $B(T_S)$ represents the blackbody radiation radiance; $T_S$ represents the real surface temperature; $\varepsilon$ represents the transmittance of the atmosphere in the thermal infrared band; $T_S$ is the temperature at the sensor, $K_1 = 0774.89 \text{ w}/(m^2 * sr * um)$, $K_2 = 1321.08$ K.

### 2.3.4. Dryness Index

The dryness index is generally expressed by the bare soil index (SI) and the index-based built-up index (IBI), which depends on the impact of constructed land on urban dryness [33]. The extensive built-up area in Panzhihua City affects the overall ecological dryness [34]. There are also many bare soil areas in the high mountains and deep valleys in Panzhihua City, which also affect the dryness factor. Therefore, a synthesis of IBI and SI should be used as dryness indexes. Considering the research methods of other scholars, the dryness index (NDBSI) of this thesis divides the weights of SI and IBI into two. The formula is as follows:

$$IBI = \frac{2 \times \frac{\rho_{mir}}{\rho_{mir}+\rho_{nir}} - \left[\frac{\rho_{nir}}{\rho_{nir}+\rho_{red}} + \frac{\rho_{green}}{\rho_{green}+\rho_{mir}}\right]}{2 \times \frac{\rho_{mir}}{\rho_{mir}+\rho_{nir}} + \left[\frac{\rho_{nir}}{\rho_{nir}+\rho_{red}} + \frac{\rho_{green}}{\rho_{green}+\rho_{mir}}\right]} \tag{12}$$

$$SI = \frac{[(\rho_{mir} + \rho_{red}) - (\rho_{nir} + \rho_{blue})]}{[(\rho_{nir} + \rho_{red})] + [(\rho_{nir} + \rho_{blue})]} \tag{13}$$

$$NDBSI = \frac{IBI + SI}{2} \tag{14}$$

where: $\rho_{mir}, \rho_{nir}, \rho_{red}, \rho_{green}, \rho_{blue}$ represents the corresponding reflectance of mid-infrared band, near-infrared band, red band, green band, and blue band, respectively.

### 2.4. Principal Component Analysis

The four index factors extracted were normalized first and then analyzed by principal component analysis. Its normalized formula is as follows:

$$P\theta_i = \frac{\theta_i - \theta_{min}}{\theta_{max} - \theta_{min}} \tag{15}$$

where $P\theta_i$ represents a normalized index value, $\theta_i$ represents the value of this index in the pixel $i$, $\theta_{max}$ represents the maximum value of this index, $\theta_{min}$ represents the minimum value of this index.

### 2.5. Grey Relational Analysis

In this study, we used the grey relational analysis model created by Professor Deng Julong to analyze these seven factors of Panzhihua City [35], and calculated the correlation coefficients $y_{xi}(t)$ of each comparison factor and the reference factor from the data obtained after the processing [30]. Its normalized formula is as follows:

$$y_{xi}(t) = \frac{\underset{i}{min}\,\underset{t}{min}\,|X_0(t) - X_i(t)| + \rho\,\underset{i}{max}\,\underset{t}{max}\,|X_0(t) - X_i(t)|}{|X_0(t) - X_i(t)| + \rho\,\underset{i}{max}\,\underset{t}{max}\,|X_0(t) - X_i(t)|} \quad (16)$$

where $\rho$ is the distinguishing coefficient; to avoid large differences in the calculation process, which may cause the data to be inaccurate, its value is usually set to 0; $|X_0(t) - X_i(t)|$ is the absolute difference between the two factors; $\underset{i}{min}\,\underset{t}{min}\,|X_0(t) - X_i(t)|$ is the minimum value of the difference; $\underset{i}{max}\,\underset{t}{max}\,|X_0(t) - X_i(t)|$ is the maximum value of the difference.

## 3. Results

### 3.1. Results of Ecological Factor Index Extraction

By extracting and analyzing the image data of Panzhihua City in 2000, 2005, 2010, 2015 and 2020, we obtained the four RSEI indicators of humidity, greenness, heat, and dryness in Panzhihua City for five periods in 2000, 2005, 2010, 2015 and 2020. The maximum, minimum, average, and standard deviation of the four indexes was calculated in each period [36], quantitatively and qualitatively analyzing the distribution of the four indexes in Panzhihua City in the 20 years of data.

As shown in Figure 2(a1–a5), NDVI of Panzhihua city has increased significantly from 2000 to 2020, especially from 2010 to 2015. The vegetation in the Eastern and western regions has increased significantly, and the areas with high NDVI in Yanbian County and Miyi County are also growing. As shown in Figure 2(b1–b5), the value of WET decreased from 2000 to 2005 in the study area, while it turned to increase from 2005 to 2020. However, the WET values of western Yanbian County changed little and were always low. The WET values in the Middle of Dong Area have been increasing since 2000. It can be seen from Figure 2(c1–c5) that the land surface temperature in areas with more vegetation is lower. The value of LST in the middle of Yanbian County decreased from 2000 to 2010 and then began to rise, with a trend of continuous rise. After rising in 2005, the land surface temperature of Miyi County decreased from 2005 to 2010, but then continued to rise again. From 2015 to 2020, the overall LST showed a downward trend. Figure 2(d1–d5) shows the change of NDBSI value from 2000 to 2020. It can be seen that the NDBSI value is always high in the eastern part of Renhe District due to the lack of vegetation. The NDBSI value of Xi Area and Dong Area is higher overall. However, the NDBSI value of Yanbian County and Miyi County is declining continuously, which is inseparable from the proposal of relevant policies such as returning farmland to forest and grassland.

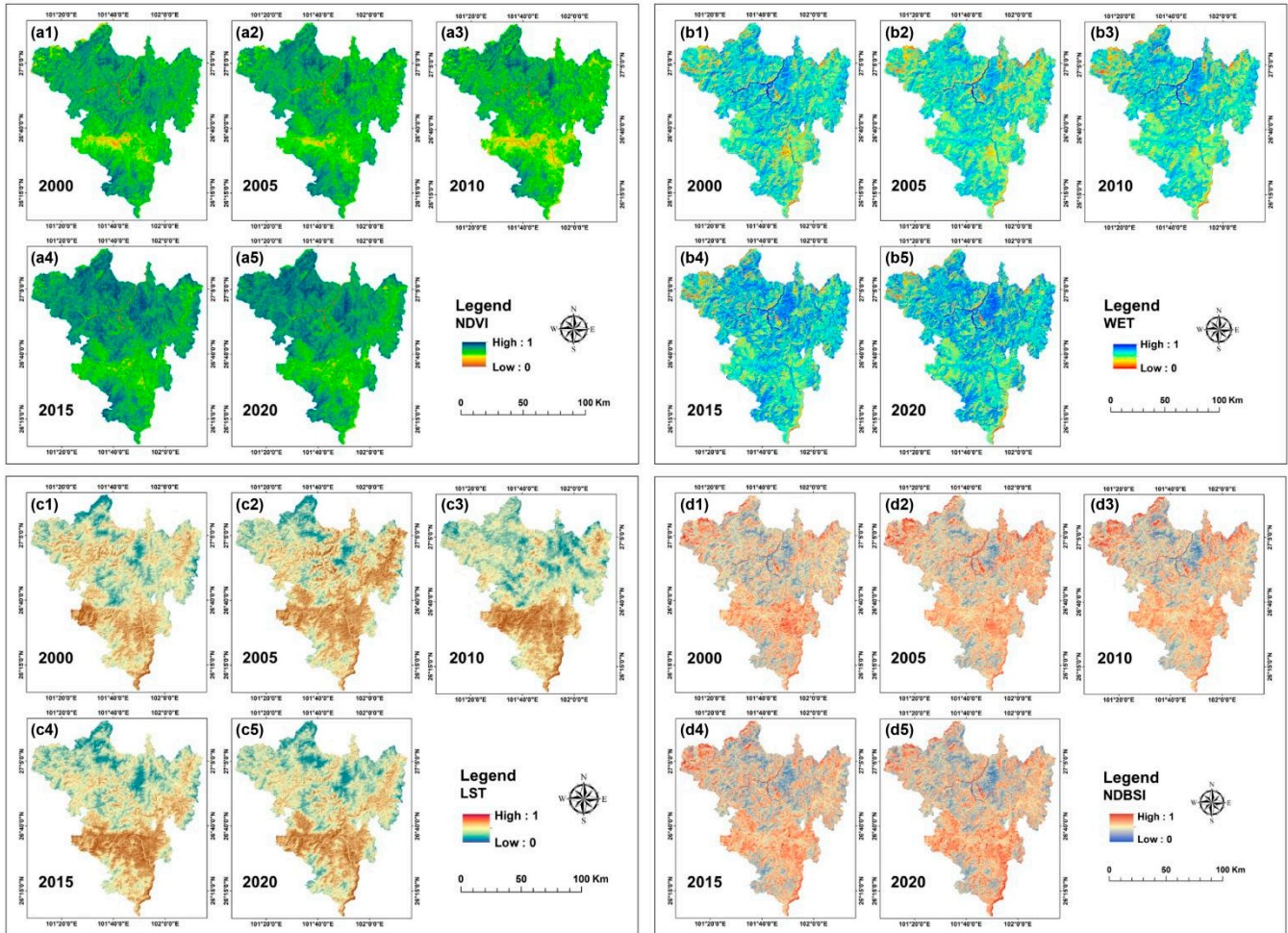

**Figure 2.** The distribution of ecological factor index in Panzhihua City in 2000-2020. Note: (**a1–a5**): NDVI distribution in 2000, 2005, 2010, 2015, and 2020; (**b1–b5**): WET distribution in 2000, 2005, 2010, 2015, and 2020; (**c1–c5**): LST distribution in 2000, 2005, 2010, 2015, and 2020; (**d1–d5**): NDBSI distribution in 2000, 2005, 2010, 2015, and 2020.

### 3.2. Principal Component Analysis Results

After normalizing the greenness, humidity, dryness, and heat indexes of the five periods of 2000, 2005, 2010, 2015, and 2020, the principal components were extracted to derive the RSEI of each period [37]. Table 2 shows the eigenvalue and eigenvalue contribution rate of each period of Panzhihua City.

**Table 2.** Statistical table of characteristic values of Panzhihua City in each period.

| Year | 2000 | | 2005 | | 2010 | | 2015 | | 2020 | |
|---|---|---|---|---|---|---|---|---|---|---|
| Principal Component | Eigenvalue | Eigenvalue Contribution Rate (%) | Eigenvalue | Eigenvalue Contribution Rate (%) | Eigenvalue | Eigenvalue Contribution Rate (%) | Eigenvalue | Eigenvalue Contribution Rate (%) | Eigenvalue | Eigenvalue Contribution Rate (%) |
| PC1 | 0.0370 | 74.54 | 0.0352 | 76.09 | 0.0448 | 81.78 | 0.0463 | 79.18 | 0.0473 | 83.43 |
| PC2 | 0.0110 | 19.34 | 0.0107 | 20.19 | 0.0118 | 10.47 | 0.0102 | 11.64 | 0.0119 | 9.97 |
| PC3 | 0.0088 | 5.40 | 0.0067 | 2.57 | 0.0066 | 6.30 | 0.0079 | 8.45 | 0.0064 | 5.76 |
| PC4 | 0.0004 | 0.72 | 0.0006 | 1.15 | 0.0009 | 1.45 | 0.0004 | 0.73 | 0.0005 | 0.84 |

The statistical results show that the eigenvalue contribution rate of the first principal component in these five periods was higher than 74%, indicating that the synthesized PC1 already contained most of the information on greenness, humidity, dryness, and heat and was used as the index to evaluate the environmental quality of Panzhihua City [38].

Table 3 lists each index feature vector of Panzhihua City in each period. The PC1 values of NDVI and WET in each period are positive, while the PC1 of LST and NDSI is negative. It indicates that NDVI and WET played a positive role in the evaluation of ecological quality, while LST and NDSI played a negative role.

**Table 3.** Characteristic vector statistical table of four indexes in each period of Panzhihua City.

| Year | Index | PC1 | PC2 | PC3 | PC4 |
|------|-------|------|------|------|------|
| 2000 | NDVI | 0.2993 | 0.2196 | −0.2849 | −0.8836 |
| | WET | 0.3251 | −0.9209 | 0.1392 | −0.1637 |
| | NDBSI | −0.6756 | −0.0716 | 0.5895 | 0.4368 |
| | LST | −0.5901 | −0.3138 | −0.7428 | −0.0383 |
| 2005 | NDVI | 0.4516 | 0.5562 | −0.2131 | −0.6641 |
| | WET | 0.5146 | −0.7703 | −03245 | 0.1909 |
| | NDBSI | −0.5288 | −0.3096 | −0.3096 | −0.7222 |
| | LST | −0.5014 | 0.0371 | 0.0371 | −0.3275 |
| 2010 | NDVI | 0.5423 | 0.4321 | 0.0155 | −0.7203 |
| | WET | 0.3064 | −0.8086 | −0.4273 | 0.2636 |
| | NDBSI | −0.7207 | −0.1714 | 0.2032 | 0.6407 |
| | LST | −0.3052 | 0.3603 | −0.8808 | −0.0326 |
| 2015 | NDVI | 0.3052 | 0.4549 | −0.0410 | −0.8355 |
| | WET | 0.4964 | −0.6571 | −0.5471 | −0.1495 |
| | NDBSI | −0.6583 | −0.5121 | 0.1618 | −0.5272 |
| | LST | −0.4763 | 0.3144 | −0.8202 | 0.0375 |
| 2020 | NDVI | 0.3022 | 0.3939 | −0.1988 | 0.8449 |
| | WET | 0.5075 | −0.8081 | −0.2679 | −0.1321 |
| | NDBSI | −0.6554 | −0.4376 | 0.3337 | −0.5170 |
| | LST | −0.4705 | −0.0089 | −0.8816 | 0.03496 |

### 3.3. Analysis of Environmental Quality Characteristics

Through the above principal component analysis of the four indicators of 2000, 2005, 2010, 2015, and 2020, the RSEI was obtained, which could further help to quantitatively analyze the factors affecting the environmental change of Panzhihua City.

In order to better refine the distribution of environmental quality and make a clear evaluation, we divided the RSEI index of each period into five equal levels (excellent, good, moderate, fair, and poor) [39], with a spacing of 0.2 for each level (Table 4) [38].

**Table 4.** Remote sensing ecological index classification table.

| Level Index | Feature Description | Grade |
|-------------|---------------------|-------|
| Poor | The vegetation cover is low and the ground is arid, the ground is rocky and leaky. Human life is greatly restricted by the environment, and the quality of the environment is very bad. | [0–0.2] |
| Fair | The vegetation cover is low and the ground is less rainy and arid with fewer species. Human life is significantly affected by the environment, and the quality of the environment is poor. | [0.2–0.4] |
| Moderate | The vegetation coverage is general and the biodiversity is moderate. Human life is generally disturbed by the environment, and the quality of the environment is medium. | [0.4–0.6] |
| Good | The vegetation coverage is high and the climate is humid. Biodiversity is rich, and the environment is helpful to human life. | [0.6–0.8] |
| Excellent | The vegetation coverage is high and soil organic matter is rich. Environmental quality is very high, and very suitable for humans living life. | [0.8–1] |

Figure 3 shows the distribution of RSEI levels in Panzhihua City in each period. The "moderate" areas in each period were the most abundant, indicating that the environmental quality of Panzhihua City was mostly at a general level. From 2000 to 2020, there were increasingly more "good" and "excellent" areas, which indicated that the environmental quality of Panzhihua City was improving.

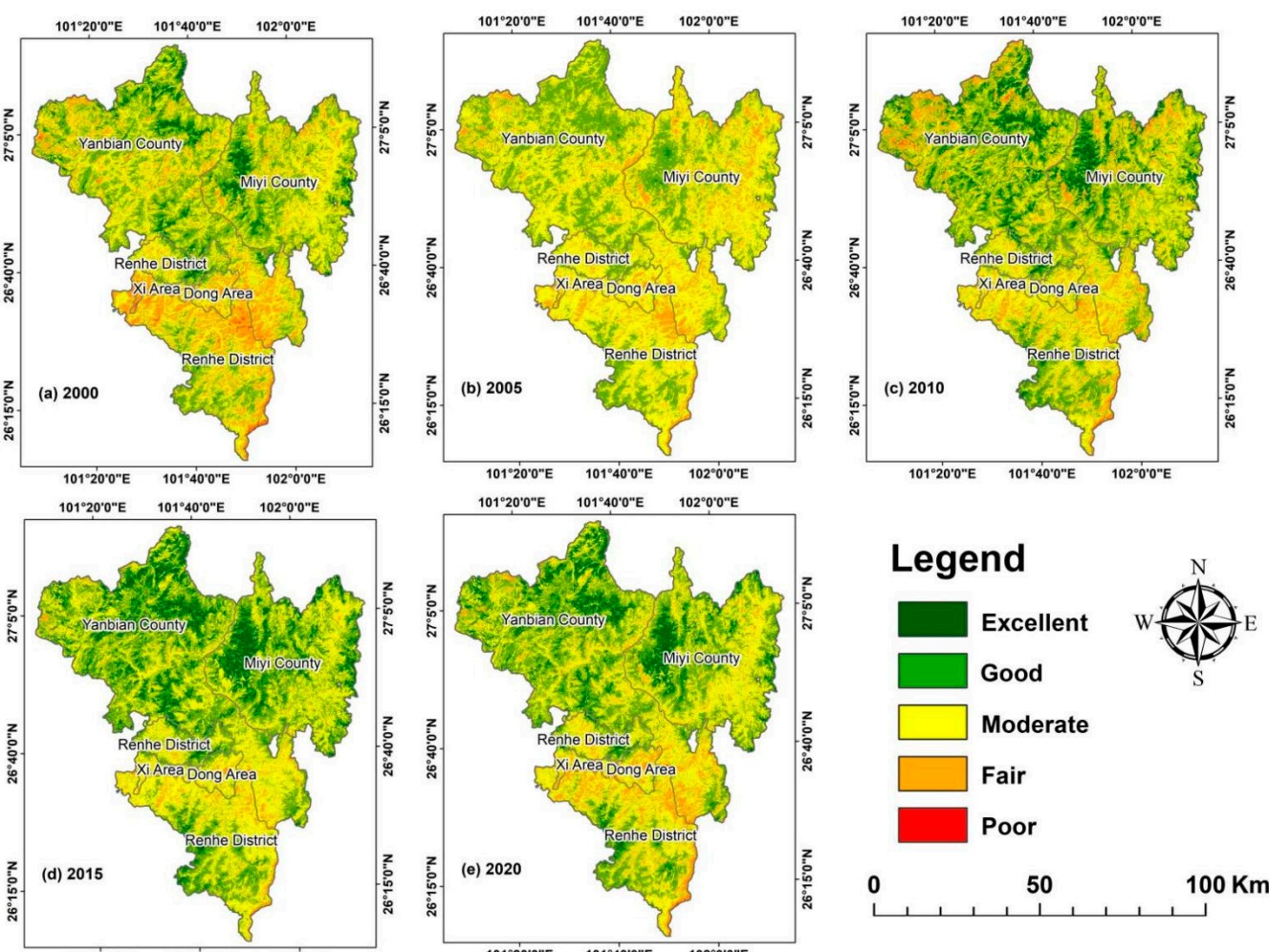

**Figure 3.** Remote sensing ecological index grade distribution map of Panzhihua City in (**a**) 2000, (**b**) 2005, (**c**) 2010, (**d**) 2015, and (**e**) 2020. Note: Classification criteria: Excellent (0.8–1); Good (0.6–0.8); Moderate (0.4–0.6); Fair (0.2–0.4); Poor (0–0.2).

The "good" and "excellent" areas mostly appear in Miyi County and Yanbian County in the north of Panzhihua City. In particular, the "good" areas in Miyi County are increasing in each period, indicating that the environment in Miyi County is improving at the fastest rate among the five districts and counties of Panzhihua City. The "poor" areas occupy the least area, and most of them are located in the southern area of Panzhihua City, indicating that the environmental quality of Panzhihua was better in the northern than in the southern area; the areas with "fair" environmental quality is small; the distribution of "poor" areas is very relevant to the eastern and western areas of Panzhihua city, where the population is concentrated.

Figure 4 shows the statistics of five RSEI evaluation levels in Panzhihua City in the past 20 years. It can be clearly seen from the figure that the RSEI levels in Panzhihua City have fluctuated from 2000 to 2020. The proportion of 'moderate' area is the largest among the five grades, and its trend was to rise first and then decline, peaking in 2005; the black lines with the evaluation grade of 'poor' are below the figure, and their area proportion is

almost negligible, indicating that there is almost no area with poor environmental quality in Panzhihua City; the RSEI of 'fair' shows a downward trend, while the RSEI of 'good' shows an overall upward trend. In summary, it can be seen that the environmental quality of Panzhihua City improved over time.

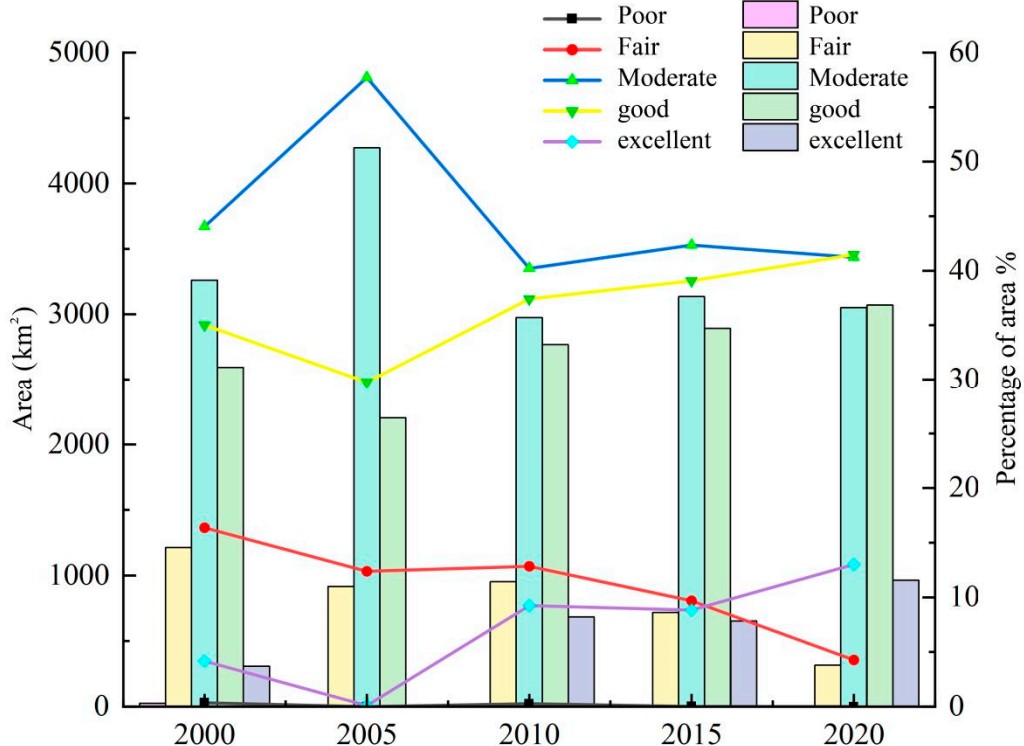

**Figure 4.** Statistical chart of the change of RSEI evaluation grade in Panzhihua City.

### 3.4. Impact of Land Use Classification on the Quality of Environmental

We selected the training samples from the remote sensing images of Panzhihua City with the help of the GEE platform, and used the support vector machine to classify land use [40]. Support vector machines have higher accuracy and less sample requirement compared to other classification methods. It overcomes the problems of overfitting and local minimum of traditional methods, and has strong generalization ability [26]. The classification accuracy had a Kappa coefficient of 0.86 and an OA coefficient of 0.89 [41]. We extracted the land use classification maps of Panzhihua City in 2000, 2005, 2010, 2015 and 2020. As shown in Figure 5, according to the current situation and research of land use investigation and related classification standards, combined with support vector machine (SVM) method and visual interpretation, the land use types interpreted by remote sensing were divided into six categories: cropland, forest, grassland, water, construction land, unutilized land [42].

In order to facilitate the statistics of the specific impact of different land-use types on environmental quality (Table A4), the average RSEI of each land-use type was calculated. Overall, the RSEI values of six land use types in Panzhihua City have improved, indicating that the environmental quality has effectively improved. As shown in Table 5, the average RSEI of cropland increased from 0.559 to 0.653 from 2000 to 2020, with a growth rate of 16.8%, which obviously promoted the environmental quality of all Panzhihua City. The proportion of cropland area in Panzhihua City increased from 20.19% to 25.92% in the past 20 years; the RSEI also gradually increased, confirming an improvement in the environmental quality of Panzhihua City. Among the six land use types, the average RSEI of the water area is the smallest, and the water area of Panzhihua City has not changed greatly, with two major watersheds, Yalong River and Jinsha River, indicating that the

watershed has an inhibitory effect on the environmental quality of Panzhihua City. The average RSEI of forest increased from 0.556 to 0.659 from 2000 to 2020, with a growth rate of 18.55%, while the forest area also decreased from 70.63% to 65.16%. The proportion of forest area in Panzhihua City reached more than half, which promoted the environmental quality in the study area (Tables A1–A4).

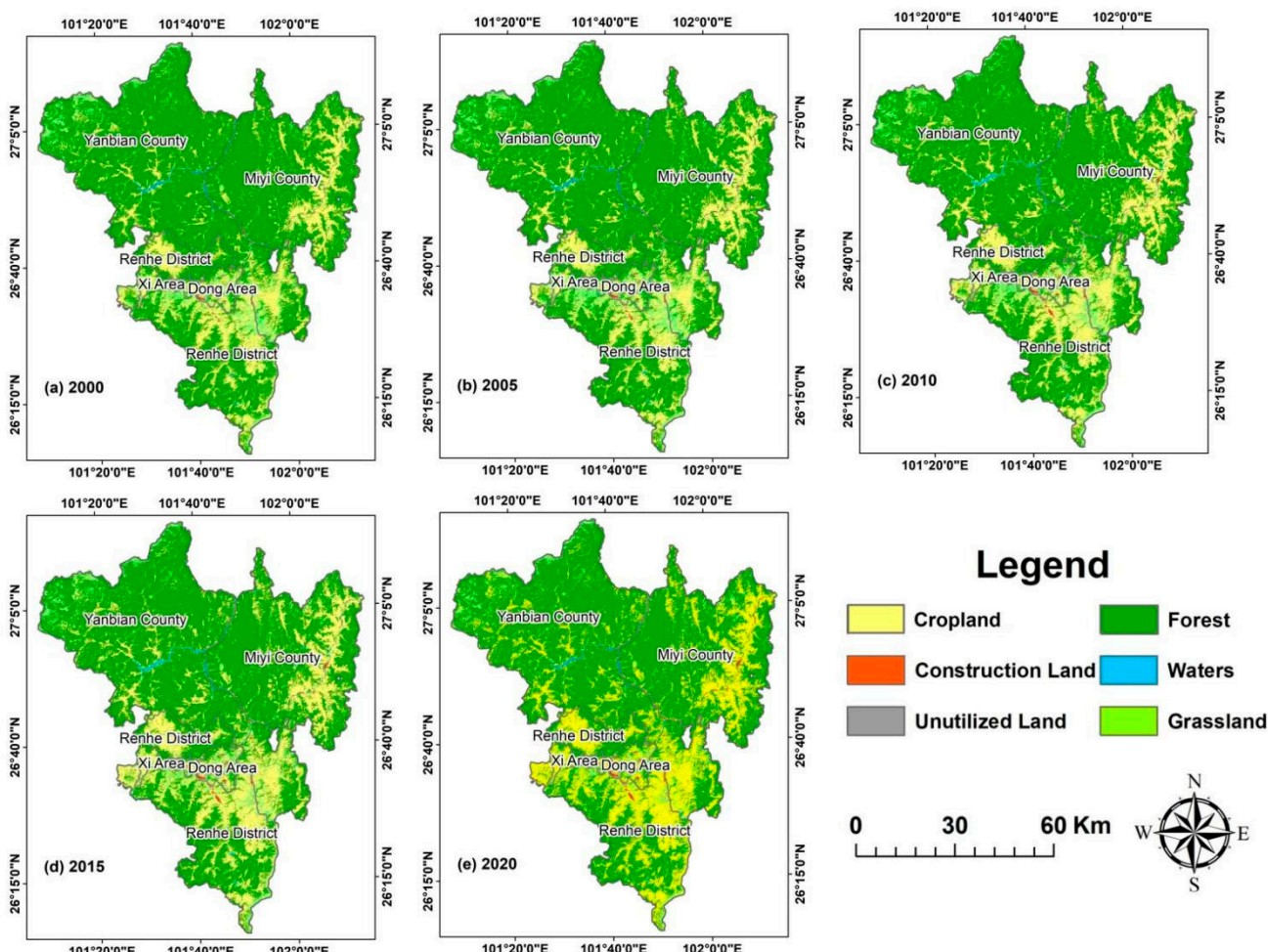

**Figure 5.** Land use classification map of Panzhihua City from 2000 to 2020. Note: (**a**): 2000; (**b**): 2005; (**c**): 2010; (**d**): 2015; (**e**): 2020.

**Table 5.** RSEI Change of Different Land Use Types in Panzhihua City from 2015 to 2020.

| Land Use Type | 2000 | 2005 | 2010 | 2015 | 2020 |
|---|---|---|---|---|---|
| Cropland | 0.559 | 0.531 | 0.587 | 0.624 | 0.653 |
| Forest | 0.556 | 0.545 | 0.582 | 0.621 | 0.659 |
| Grassland | 0.571 | 0.543 | 0.591 | 0.609 | 0.654 |
| Construction land | 0.547 | 0.509 | 0.580 | 0.591 | 0.646 |
| Water | 0.488 | 0.477 | 0.499 | 0.501 | 0.508 |
| Unutilized land | 0.566 | 0.568 | 0.579 | 0.596 | 0.633 |

*3.5. Impact Analysis of the Mining Area on Environmental Quality*

In this study, the Buffer module of ArcGIS software was used to establish buffer zones with a radius of 2 km, 3 km, and 5 km, respectively, with the mining rights points as the center. We overlaid the generated buffer zones onto the existing remotely sensed ecological index data and analyzed the changes in ecological quality within the buffer zones by the time series. Figure 6 shows the environmental quality distribution map of Panzhihua

Mining Rights Center of 2010 as an example. There are twenty-eight coal mines in Miyi County, Yanbian County, and the eastern and western districts of Panzhihua City, which are relatively close to areas of intense human activities. The impact of mines on urban environmental quality was analyzed qualitatively and quantitatively in combination with the RSEI distribution map of Panzhihua City.

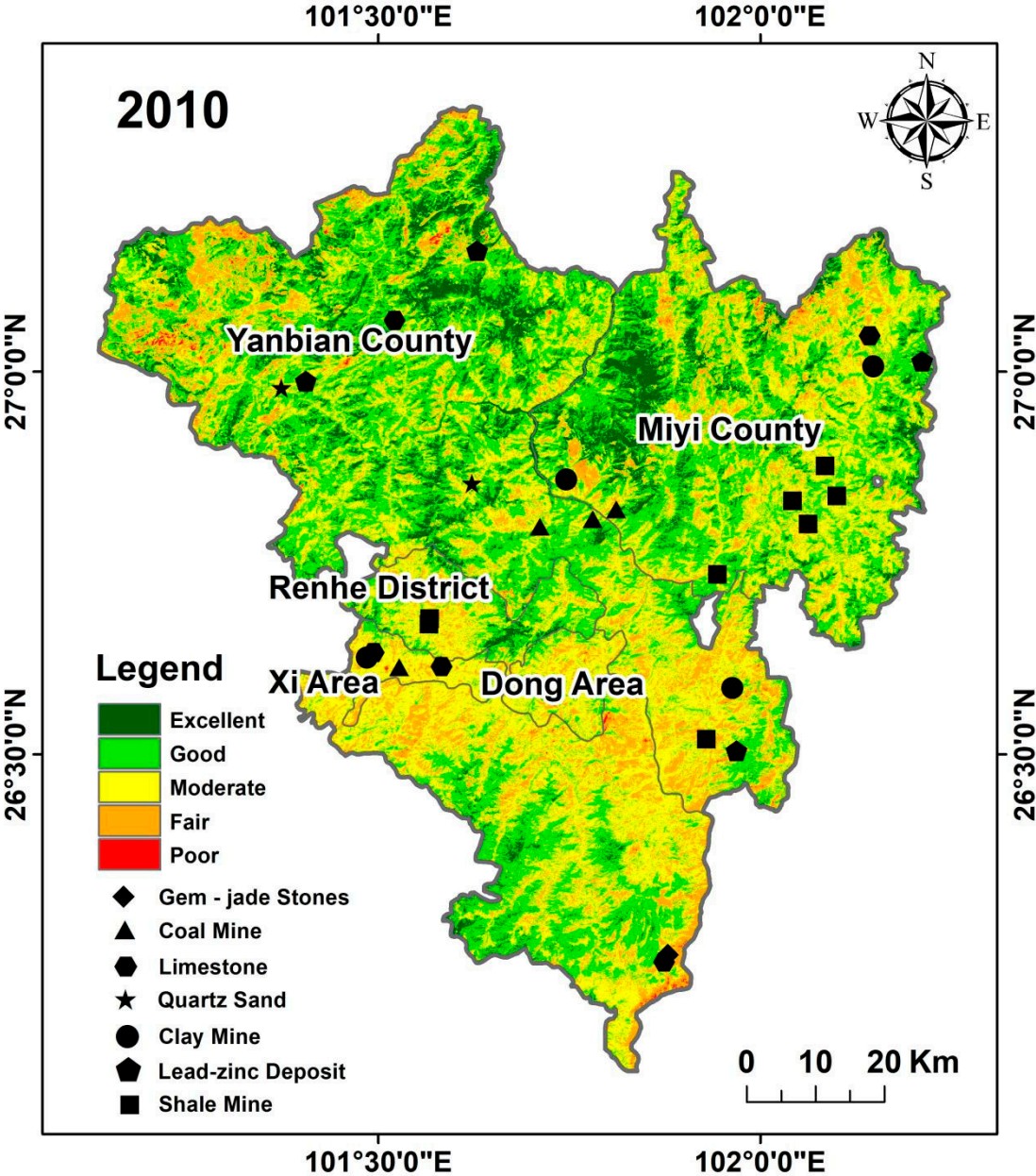

**Figure 6.** Distribution of mining rights center points in Panzhihua City. Note: Classification criteria: Excellent (0.8–1); Good (0.6–0.8); Moderate (0.4–0.6); Fair (0.2–0.4); Poor (0–0.2).

Figure 7a shows that the mining area has an impact on the environmental quality of the surrounding 2 km. Most of the area has an environmental grade quality of 'fair' and 'moderate' because metals such as iron and cobalt around the mining area stunt vegetation growth, with patches of bare land that keep humidity relatively low.

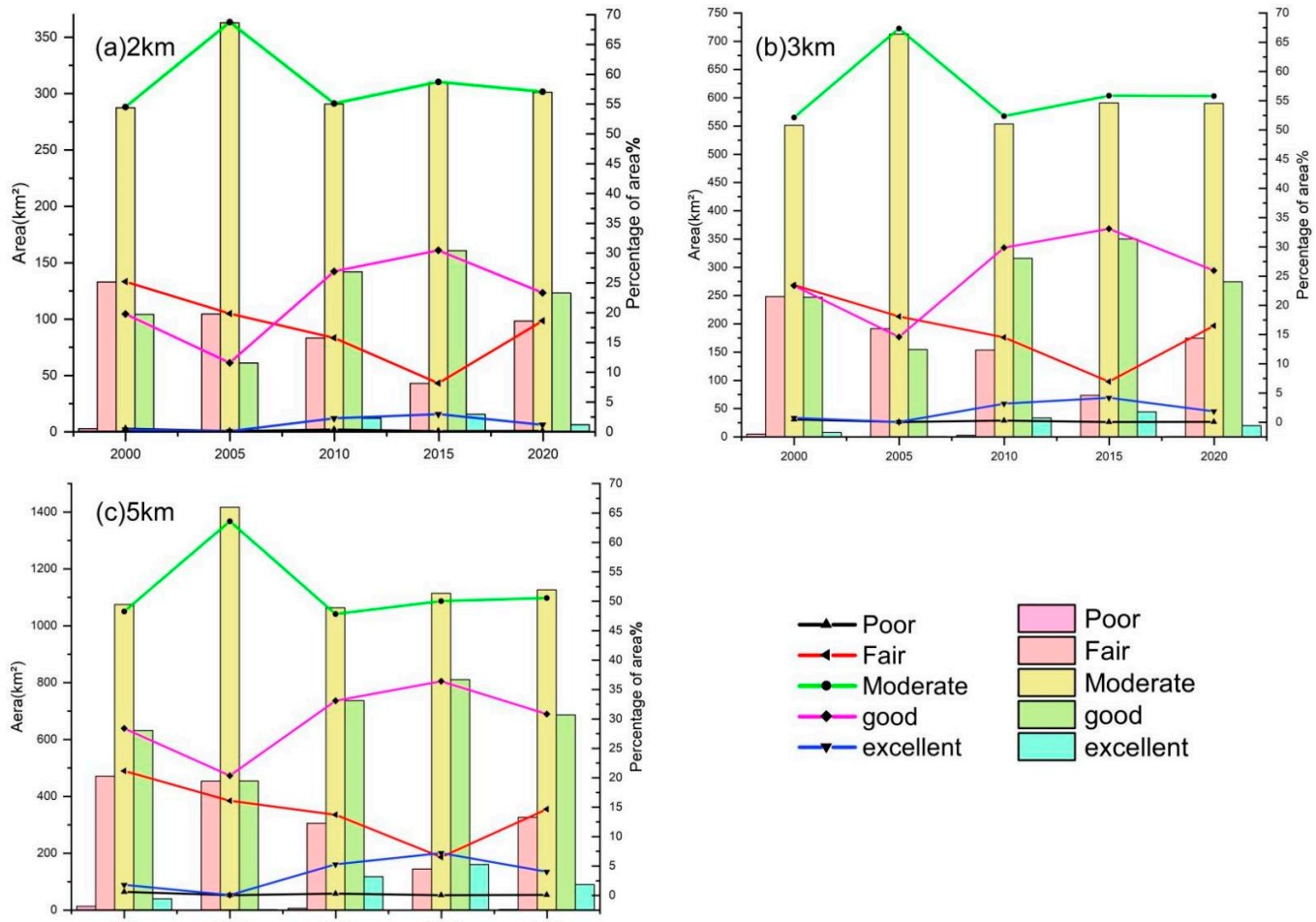

**Figure 7.** Remote sensing ecological index in 2, 3, 5 km buffer zone of Panzhihua mining rights center. Note: (**a**): 2 km; (**b**): 3 km; (**c**): 5 km.

A 3-km mining buffer zone to quantitatively was established to analyze the impact of mining areas on environmental quality. Figure 7b shows that the environmental quality grade is still dominated by the "moderate" area, accounting for almost more than half of the total area, while the area of "excellent" environmental quality is 7.31 km², 0.11 km², 33.18 km², 43.76 km² and 19.34 km² in order of year, showing an overall trend of decreasing and then increasing; the grade of "good" shows a trend first decreasing and then increasing, with little total area change.

In order to further and quantitatively study the impact of the mining area on the quality of the surrounding environment, a buffer zone with a 5-km radius from the center of the mining rights rights was established again. From Figure 7c, it can be seen that the percentages of areas with "good" environmental quality are 28.35%, 20.33%, 33.05%, 36.37%, and 30.79% from 2000 to 2020; there was an overall downward trend and then a slow upward trend, with the least in 2005 and the highest in 2015. The areas of "fair" environmental grade were 21.11%, 16.05%, 13.67%, 6.46%, and 14.63%, respectively, showing a slow downward trend, while the area in 2015 was the least. On the contrary, the "excellent" environmental grade area peaked in 2015.

### 3.6. Impact Analysis of Mining Methods on Environmental Quality

To further study the impact of open-pit mining, underground mining and non-mine mining on the environmental quality of Panzhihua City, we selected typical mining areas (Figure 8).

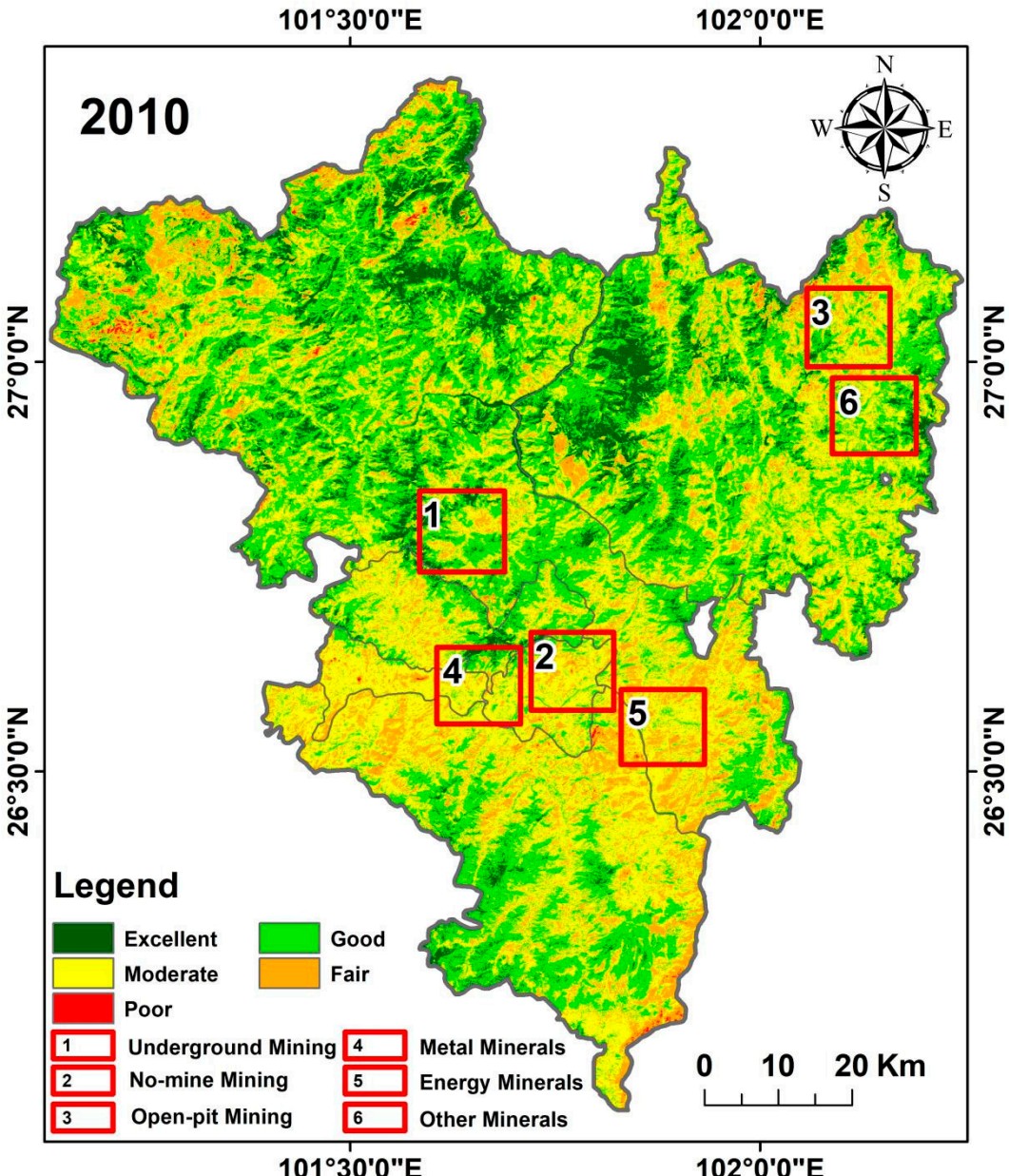

**Figure 8.** Location of typical mining areas in Panzhihua City. Note: Classification criteria: Excellent (0.8–1); Good (0.6–0.8); Moderate (0.4–0.6); Fair (0.2–0.4); Poor (0–0.2).

Figure 9 shows underground mining in the northern part of Yanbian County. The deposit is buried a little deeper from the surface, so it is necessary to dig a tunnel when mining the deposit to avoid the direct influence of vegetation and land use on the ground in the process of mining; thus, most of the figure shows green and the area of 'excellent' and 'good' environmental quality is the majority. With the increase of time, the "excellent" and "good" areas became the largest in 2015. The environmental quality generally shows the situation of decreasing first and then increasing, according to the buffer zone distribution. The environmental quality of the 5-km buffer is better than the 3-km buffer, and the ecological quality within 2 km is the worst, which indicates that the mining influence range of the mine shows a radiation-like trend.

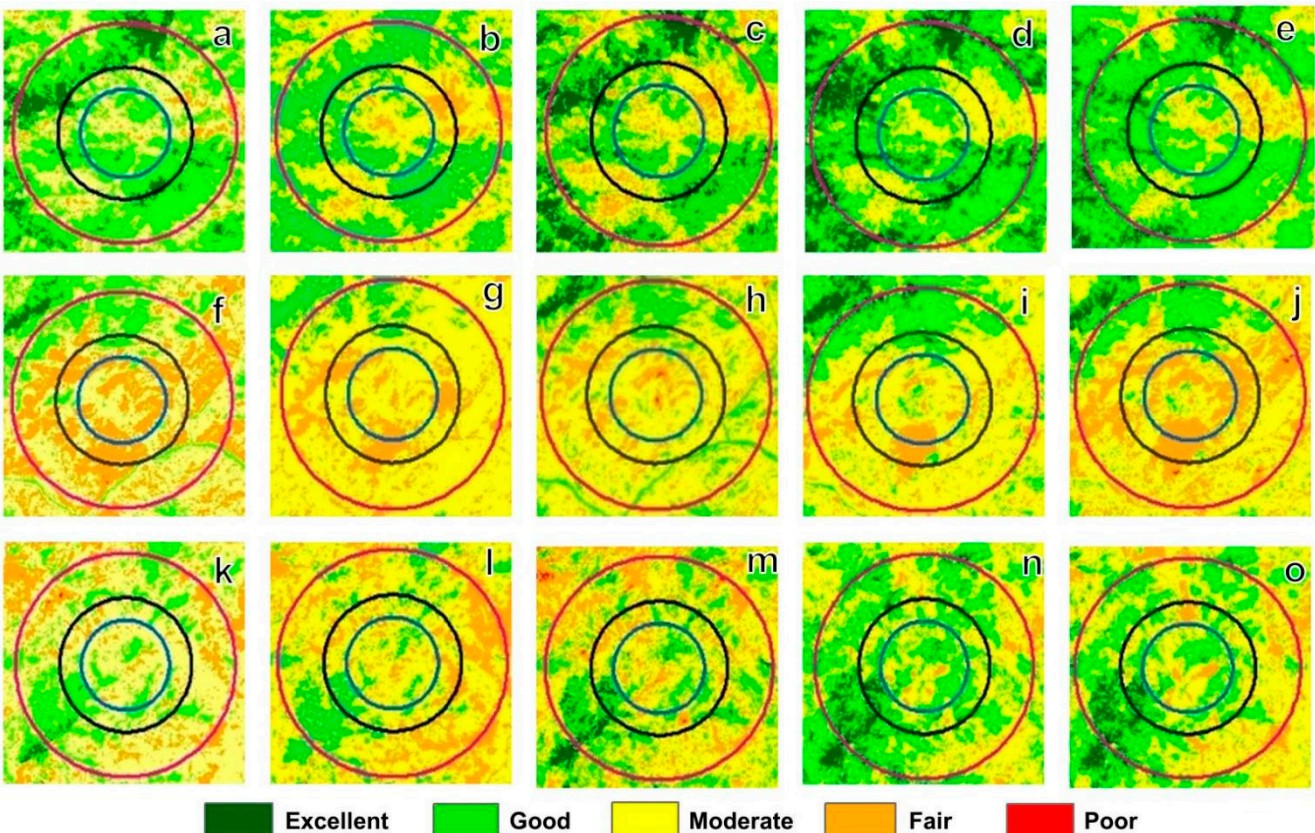

**Figure 9.** Distribution of (**a–e**) underground mining, (**f–j**) no-mine mining and (**k–o**) open-pit mining buffer zone in Panzhihua City in 2000, 2005, 2010, 2015 and 2020.

Labels 1, 2, and 3 are the locations of underground mining, non-mine mining, and open-pit mining areas, which are made as the buffers with 2, 3, and 5 km, respectively.

*3.7. Analysis of the Impact of Different Minerals on Environmental Quality*

To further study the impact of different minerals (metal, energy and other minerals) on the quality of the environment, mining areas with typical characteristics were selected for further study. In Figure 8, the labels 4, 5 and 6 are the locations of metal mines, energy mines and other minerals, and 2-km, 3-km and 5-km buffer zones, respectively.

It can be seen from Figure 10a–e that the development and utilization of metal mines (mostly vanadium-titanium magnetite developed by Panzhihua Iron and Steel (Group) Co. is relatively high. The majority of the map is in the "fair" and the "moderate", indicating that the ecological quality of this area is in urgent need of improvement. It shows a gradual improvement over time cycle. In Figure 10f–j, the ecological quality of energy mines such as coal mines is slightly better than that of metal mines, especially when the buffer zone is 5 km, the 'good' ecological grade appears, while the grade in the 2-km buffer zone is still very poor. Since the scale of mining of other mineral species is lower than that of metal and energy mines, the environmental impact caused is also relatively smaller, as shown by the higher proportion of "good" areas.

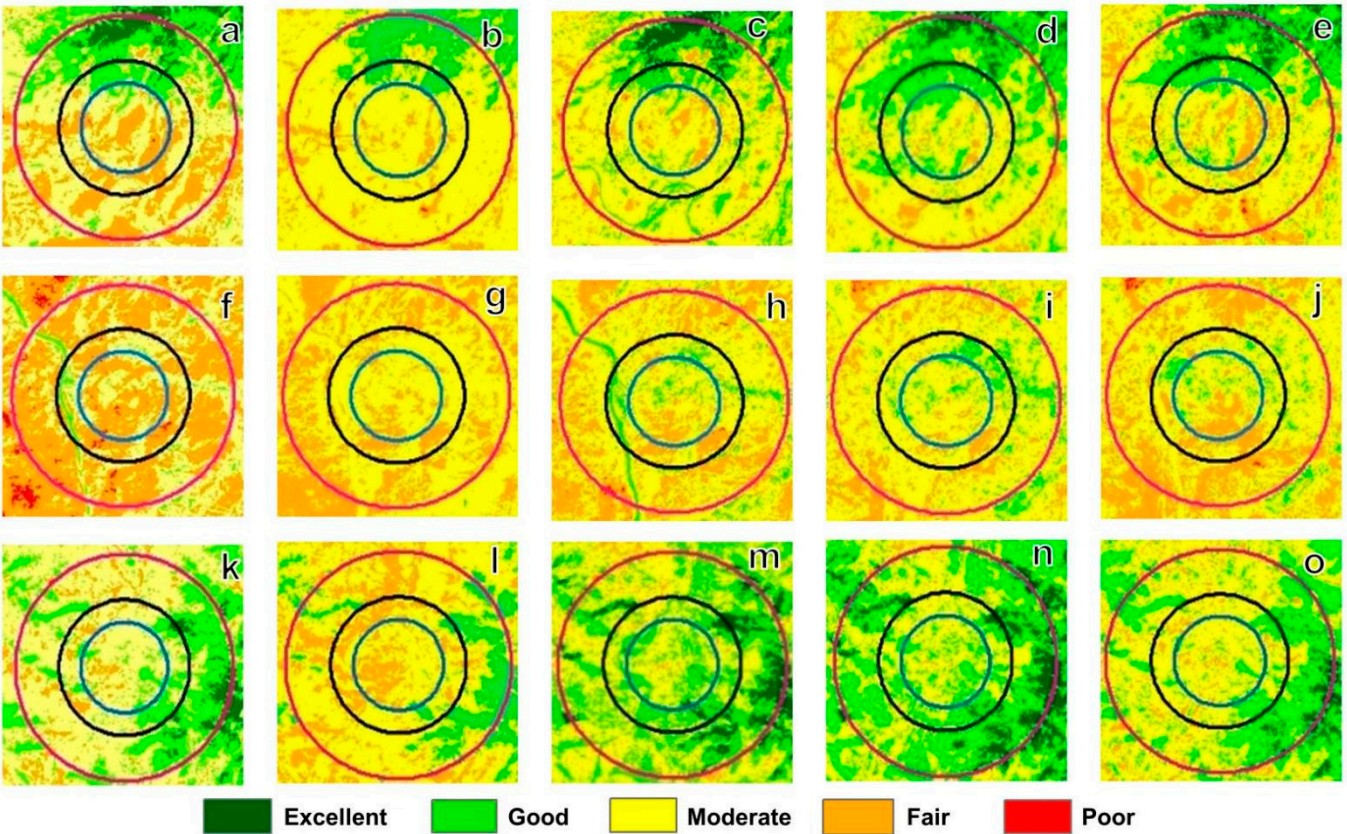

**Figure 10.** Distribution of (**a–e**) metal mine, (**f–j**) energy mine and (**k–o**) other mine buffer zones in Panzhihua City in 2000, 2005, 2010, 2015 and 2020.

After extracting the above seven factors from the statistical yearbook, we numbered them as average annual rainfall X1, total sunshine hours X2, cultivated land area X3, built-up area X4, gross product of primary industry X5, gross product of secondary industry X6 and gross product of tertiary industry X7 and regard them as comparative factors. Then, we use the remote sensing ecological index X0 obtained in Section 4 as a reference factor. To facilitate subsequent statistics and analysis and to eliminate the unit and data size inconsistencies between different factors, we standardized them and then calculated the correlation coefficient $y_{xi}(t)$ between each comparison factor and the reference factor for the processed data. In order to calculate the correlation between factors from 2000 to 2020, we first calculated the mean to unify the data results of the five periods and then calculated the correlation between each factor and the remote sensing ecological index (Figure 11).

These seven correlation values are all greater than 0.5, indicating that there is a significant correlation with the RSEI value. The highest correlation value is for the meteorological factors. The average annual rainfall and total sunshine hours are 0.83 and 0.82, respectively, which indicates that the meteorological conditions of Panzhihua City, which are greatly related to the geographical location and climate, have a powerful influence on environmental quality. Sufficient sunshine and rainfall in summer and autumn provide the necessary conditions for the growth of vegetation, and therefore the vegetation cover is relatively high, which has a positive impact on environmental quality. The ranking of these seven influencing factors in descending order of correlation is annual average rainfall > total sunshine hours > cultivated land area > built-up area > gross product of secondary industry > gross product of primary industry > gross product of tertiary industry.

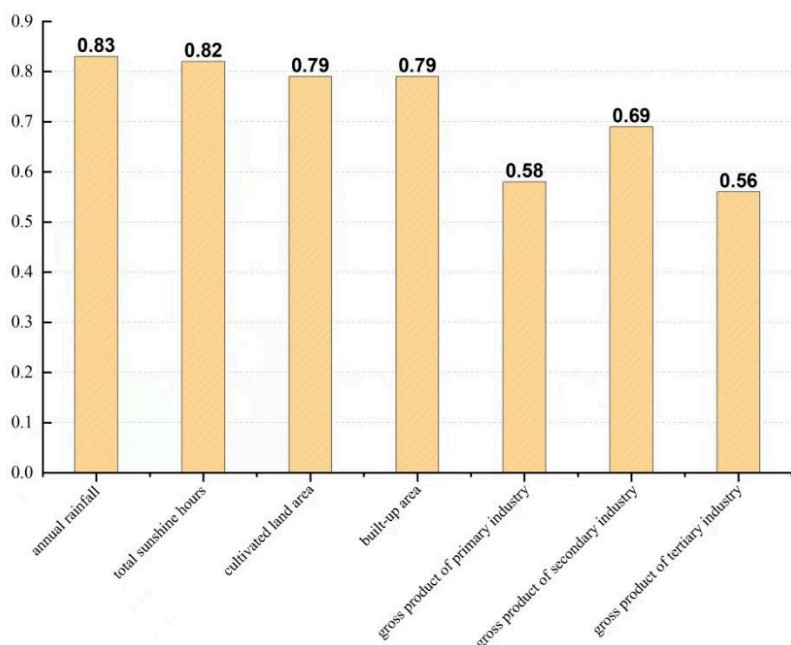

**Figure 11.** Gray correlation analysis of influencing factors.

## 4. Discussion

### 4.1. Impact Factor Analysis

The data results showed that the environmental quality index of Panzhihua City from 2000 to 2020 first decreased and then increased. The overall trend has been steadily increasing since 2005. The average RSEI increased by 0.0747 in the past 20 years, with a growth rate of 13.56%, indicating that the environmental quality of Panzhihua City has improved from 2000 to 2020. As an important resource and industrial city in Sichuan Province, Pangzhihua has rich mining resources. Its environmental quality was damaged to different degrees as a result of coal and steel production more than ten years ago [43]. Then, with the transformation of the urban economy, the traditional "steel city" has been transformed into the "Sunshine Flower City" of China using light and heat resources, and the ecological quality has steadily improved [44].

The change in land use indirectly changed the environmental factors in the region, which will have negative or positive impacts on the environmental quality of Panzhihua City. We generally classify cropland, forest, and water as ecological land. These land types can not only prevent wind and promote sand fixation and purify air, but also maintain water in the soil, thus protecting and regulating the ecosystem [45]. As analyzed above, the increase of cropland, forest land and grassland increase vegetation cover and water and soil conservation capacity, which improve regional environmental quality; on the other hand, the decrease of cropland, forest land and grassland in the region due to their transfer will worsen the environmental quality [46]. The increase in the area of construction land, unutilized land intensifies the urban heat island effect and have a negative effect on regional environmental quality. The increase in cropland area promotes an increase in soil moisture and vegetation cover, which positively impacts the whole remote sensing ecological index of Panzhihua City. However, the increase in construction land area is bound to reduce the area of cropland, forest, and grassland in the city to varying degrees, thus also impacting the environmental quality of the city [47]. Therefore, we should continue to promote urban greening and strictly control the area of the built-up to improve the quality of the urban environment. Its impact on the city's economy is also very important [48]. The growth of the total industrial value of cropland, forest land, and grassland has generated enough impetus and promotion for the urban economy. The economic growth provides the city with enough funds to continue to play a role in urban infrastructure and rationally

plan a livable and comfortable urban environment [49]. On the whole, "excellent" and "good", which represent good environmental quality, reached a low point in 2005, when they occupied a relatively low area. After the closure of the mining area in 2014 and the subsequent land reclamation and other related measures, the overall environmental quality around the mining areas in Panzhihua City improved significantly. The closure of the mine, tree-planting and reforestation increased the vegetation cover and decreased the bare soil area, changing the surrounding environmental quality. The closure of mines and other related policies, as well as the transformation of Panzhihua City from a resource-based to a tourist city in the past few years, have promoted land reclamation and positively impacted the environmental quality of the city, which is now a livable and comfortable resort city [50].

Analysis of Figure 9f–j shows that the overall environmental quality of this area is "moderate" and "good", and shows a trend of slow improvement over time. The buffer zone map shows that the environmental quality of 2 km around the mining area is worse than that of the 5 km, highlighting the impact of the mining area on the surrounding environment. In Figure 9k–o, there are a lot of yellow areas representing "moderate" environmental quality and coffee color representing "poor" environmental conditions, which indicates that open-pit mining has a greater impact on environmental quality compared with other mining methods [51]. The development and utilization of open-pit mining, most of which are developed and utilized by Panzhihua Iron and Steel (Group) Co., is relatively high. Long-term development produces waste water, waste gas and other waste materials that seriously affect the growth of surrounding vegetation and cause groundwater pollution [52]. In summary, the different mining methods have caused different degrees of harm to the quality of the environment in Panzhihua City.

### 4.2. Recommendations

The analysis shows that the environmental quality of Panzhihua City has changed to different degrees under the influence of the changing land use and the influence of numerous mines in the surrounding environment, as well as the unique topographic factors and economic aspects. Based on the existing environmental quality status, combined with Panzhihua City Statistical Yearbook over the years, we make the following recommendations for the future development of Panzhihua City:

(1) Reasonable adjustment of land-use structure. In 2020, the largest proportion of land-use types in Panzhihua City was forest, reaching 4821.45 km$^2$. However, the area of forest and grassland decreased by nearly 200 km$^2$ based on 2015. According to this trend, the area of forest and grassland will continue to decrease, which will seriously affect the environmental quality of Panzhihua City. It is necessary to increase the vegetated areas in the region, optimizing the land use pattern and layout. This will lead to an increase in the area of forest, water and grassland, improve the vegetation coverage and the greening area in the construction land area, increase the soil and water moisture and retention capacity, reasonably control the area of construction land, which is a benefit to build a green city with a more reasonable land-use structure.

(2) Strengthen urban environmental remediation. Pollution of gas emissions, environmental pollution of urban rivers and groundwater, ecological pollution of nature reserves, and urban green areas are currently present in the city. The abandoned garbage in city streets and roads, as well as the discharge of pollutants such as tailings and slags from a large number of mining companies in Panzhihua City have negatively affected the urban environment. We are supposed to establish an environmental pollution prevention mechanism to strengthen management, and regularly and irregularly carry out strict review and remediation measures to build a livable and harmonious urban environment.

(3) Vigorously develop the green economy. As a key period in the transformation of resource-based industrial cities, we should capture the current green transformation and development linkage mechanism. For the existing mineral resources, on the basis of rational development and utilization, we are expected to pay more attention to protecting the

surrounding environment, promote the continuous transformation and stable development of industrial parks and key leading enterprises in the whole city and accelerate to realize the transformation development of industrial green and stable green. Building more green areas and wetland parks and increasing the vegetation cover in roads and subdivisions within the city can contribute to rational urban planning. Changing the landscape pattern of urban green space and strengthening the construction of green space environments in old urban areas and urban-rural areas can have a positive impact on the ecological landscape pattern of the city.

(4) Strengthen the supervision of the mine ecological environment. According to the previous analysis, the minerals in Panzhihua City have a great impact on the environmental quality, as over-exploitation has caused ground subsidence, landslides and debris flow, and waste problems such as solid waste generated during mining, surface and groundwater pollution caused by wastewater discharge, air pollution from the exhaust gas and dust emission. These activities destroy the vegetation and the soil, resulting in problems such as debris [53]. We should establish a technical standard system for mine technical specifications from the source, such as land reclamation, green mines, and other related measures. In the process of mining, we should regularly and irregularly make environmental dynamic tracking monitoring and evaluation. The whole mining process should conform to the standards of rational development of resources and environmental protection. In the later stage of mine closure, environmental remediation and restoration should be completed first as well as approved by the relevant departments, and then the expert group should conduct the evaluation and acceptance work [54].

*4.3. Limitations of the Study*

(1) The remote sensing images in this study were selected from Landsat 5 and Landsat 8. On account of a certain deviation in the image acquisition time, there are differences in the cloud amount, sensors, climate, and other influences that affect the quality of the remote sensing images. Although similar time periods were selected, there is a certain deviation in the data results. Additionally, this paper only selected images at an interval of five years, which can only reflect the changes in environmental quality to a certain extent. A yearly analysis of environmental quality changes may improve the accuracy of the results.

(2) In this study, the evaluation results of environmental quality in Panzhihua City were all derived from the RSEI calculation value extracted through the coupling of four factors and without reference to other factors. This is not sufficient to measure and evaluate the current situation and development trend of environmental quality in Panzhihua City. Therefore, in future research, it will be necessary to integrate the fields of ecology, physical geography, and socio-economics to complement each other and take the influence of other factors into account. In the analysis of influencing factors, we only considered land use, topography, and mining area. In future research, the influence of meteorological, hydrological, and geological disasters should also be considered.

## 5. Conclusions

In this study, the remote sensing ecological index was used to analyze the ecological quality changes in Panzhihua city in the past 20 years. From 2000 to 2020, 54.25% of Panzhihua City showed no obvious change. The proportion of environmental quality area was 5.039%, and the area with better environmental quality accounted for 40.7%, which again confirms that the environment in Panzhihua City has been improving over the 20-year period. In the process of mineral resources exploitation, the ecological environment quality of the whole city is under great pressure. Due to non-standard development in mining areas, the surrounding ecological environment will be seriously affected. However, after the closure of some mining areas in 2014, the ecological environment quality in 2015 was significantly better than that of the surrounding area in 2010, which demonstrated the positive impact of the policy.

Combined with the land use type, terrain data, mining data, and socio-economic data of Panzhihua City, we used the grey correlation degree, buffer zone, and other methods to quantitatively analyze the impact of natural factors and human factors on the ecological environment quality. The grey relational analysis was used to analyze the annual precipitation, total sunshine hours, cultivated land area, built-up area, and the gross product of the first, second, and third industries in Panzhihua City in the statistical yearbook. The results showed that the correlation between the average annual rainfall and the total sunshine hours was greater than 0.82, indicating that these two indicators have a great impact on the environmental quality of Panzhihua City.

The dynamic analysis of the multi-year remote sensing monitoring of Panzhihua is conducive to the evaluation of the dynamic development of mineral resources in the region. The results are helpful to promote the orderly exploitation and utilization of mineral resources in mining cities, and provide a layout for local economic development.

**Author Contributions:** Conceptualization, X.D. and Y.Y.; Data curation, X.D., J.Z. (Jiayun Zhou) and Z.L.; Formal analysis, X.D., W.L. (Wenyu Li), Z.L. and C.L.; Funding acquisition, Y.Y.; Investigation, X.D., Z.L. and J.Z. (Jiayun Zhou); Methodology, X.D. and H.R.; Project administration, X.D.; Resources, X.D. and Y.Y.; Software, W.L. (Weile Li) and J.Z. (Junjun Zhang); Supervision, X.D.; Validation, L.X., X.J., J.Z. (Jinbiao Zhang) and W.H.; Visualization, X.D., J.Z. (Jianwen Zeng), C.L. and Y.S.; Writing—original draft, X.D., W.L. (Wenyu Li) and C.T.; Writing—review & editing, X.D., Wenyu Li and C.T. All authors have read and agreed to the published version of the manuscript.

**Funding:** This work was supported by the Institute of Multipurpose Utilizationg of Mineral Resources, China Academy of Geological Science (Grant No. DD20221809), the Foundation of China Geological Survey (Grant No. DD20190446), the National Key Research and Development Program of China (Grant No. 2021YFC3000401), the National Natural Science Foundation of China (Grant No. 41941019), Sichuan Mineral Resources Research Center (Grant No. SCKCZY2021-ZC003), Sichuan Provincial Department of Education Humanities and Social Sciences (Zhang Daqian research) key project(Grant No. ZDQ2021-01), and Open Foundation of Sichuan Center for Disaster Economic Research(Grant No. ZHJJ2021-ZD001).

**Data Availability Statement:** The data that support the findings of this study are available from the corresponding author upon reasonable request.

**Conflicts of Interest:** No potential conflict of interest was reported by the author(s).

## Appendix A

**Table A1.** Land Use Transfer Matrix of Panzhihua City from 2000 to 2005.

| | | Land Use Types in 2005 (km$^2$) | | | | | |
| --- | --- | --- | --- | --- | --- | --- | --- |
| | | Unutilized Land | Construction Land | Waters | Grassland | Woodland | Cultivated Land |
| Land use types in 2000 (km$^2$) | Unutilized land | 1.29 | 0.46 | 0.62 | 0.19 | 0 | 0.01 |
| | Construction land | 0 | 54.57 | 0.46 | 0 | 0 | 0 |
| | Water | 0.11 | 0.42 | 68.59 | 0.64 | 0.1 | 0.57 |
| | Grassland | 0.19 | 8.27 | 0.86 | 501.65 | 26.66 | 26.74 |
| | Forest | 0 | 6.13 | 0.01 | 9.56 | 5106.57 | 110.42 |
| | Cropland | 0.01 | 5.45 | 2.58 | 109.39 | 64.33 | 1347.39 |

**Table A2.** Land Use Transfer Matrix of Panzhihua City from 2005 to 2010.

| | | Land Use Types in 2010 (km$^2$) | | | | | |
|---|---|---|---|---|---|---|---|
| | | Unutilized Land | Construction Land | Waters | Grassland | Woodland | Cultivated Land |
| Land use types in 2005 (km$^2$) | Unutilized land | 1.08 | 0.15 | 0.21 | 0 | 0 | 0.01 |
| | Construction land | 0 | 20.32 | 0.97 | 0 | 0 | 0.01 |
| | Water | 0.13 | 0.33 | 70.99 | 0.56 | 0.13 | 0.43 |
| | Grassland | 0.24 | 2.13 | 1.18 | 490.74 | 18.66 | 108.49 |
| | Forest | 0 | 0.05 | 0.01 | 14.69 | 4997.48 | 185.43 |
| | Cropland | 0.01 | 4.19 | 1.85 | 42.48 | 39.17 | 185.43 |

**Table A3.** Land Use Transfer Matrix of Panzhihua City from 2010 to 2015.

| | | Land Use Types in 2005 (km$^2$) | | | | | |
|---|---|---|---|---|---|---|---|
| | | Unutilized Land | Construction Land | Waters | Grassland | Woodland | Cultivated Land |
| Land use types in 2000 (km$^2$) | Unutilized land | 0.88 | 0.15 | 0.11 | 0.29 | 0 | 0.01 |
| | Construction land | 0 | 26.69 | 0.46 | 0 | 0 | 0 |
| | Water | 0.05 | 0.27 | 73.58 | 0.54 | 0.02 | 0.69 |
| | Grassland | 0.25 | 0.38 | 2.41 | 420.08 | 22.23 | 103.32 |
| | Forest | 0 | 0.01 | 0.01 | 16.28 | 4838.23 | 200.92 |
| | Cropland | 0.01 | 1.17 | 1.71 | 63.71 | 44.92 | 1580.29 |

**Table A4.** Land Use Transfer Matrix of Panzhihua City from 2015 to 2020.

| | | Land Use Types in 2005 (km$^2$) | | | | | |
|---|---|---|---|---|---|---|---|
| | | Unutilized Land | Construction Land | Waters | Grassland | Woodland | Cultivated Land |
| Land use types in 2000 (km$^2$) | Unutilized land | 0.48 | 0.13 | 0.17 | 0.38 | 0 | 0.01 |
| | Construction land | 0 | 28.34 | 0.33 | 0 | 0 | 0.01 |
| | Water | 0.01 | 0.20 | 75.74 | 0.88 | 0.1 | 1.46 |
| | Grassland | 0.04 | 1.25 | 0.86 | 355.31 | 20.42 | 123.04 |
| | Forest | 0 | 0.02 | 0 | 6.80 | 4743.21 | 155.36 |
| | Cropland | 0.01 | 0.75 | 1.22 | 43.29 | 57.82 | 1782.15 |

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
