# Peer review of "Effects of Mining on Urban Environmental Change: A Case Study of Panzhihua"

_remotesensing, doi:10.3390/rs14236004_

Round 1

Reviewer 1 Report

1. This article does not cover the application scenario of the selected model, whether it is suitable to be applied to Panzhihua, and also lacks the summary of previous research on environmental evaluation of Panzhihua.

2. Lines 52-54 need a citation. Line 62, “Hanqiu Xu” should be “Xu”. Line 136, URL reference is incorrect.

3. There are some minor errors, such as Line 18, "SW China" should be written in full. Line 112, “OLI”: Operational Land Imager; “MSS”: Multispectral Scanner.

4. Line 108, “figure 2-1” should be “figure 1”, line 238, the explanation of the subgraph in the title should not be missing. Please check the title of the figure in the text carefully.

The X and Y axis must be identified.

5. Data sources can use a table stating information such as time, source, type, mathematical basis, etc.

6. Line 127 Why is the SVM method used for land use classification? What is the precision, and the advantages compared to the dataset?

7. When describing the ecological quality level, the text uses a lot of "Color" to describe it, which should be replaced by the corresponding "Level Index". For example, in line 238, "The dark green and light green areas..." should be revised to "Good and excellent areas".

8. Lines 269-271, you have divided the RSEI into 5 levels, is the grading method appropriate, any reference?

9. There are some minor formatting errors, such as there should be a space before the reference number, and the formulae in the text should be centered, please check carefully.

10. The conclusion is mainly a description of the results and should reflect the innovation of the work.

Reviewer 2 Report

The article presents the environmental impact of mining in one of the Chinese regions. The authors use a series of environmental indexes and compare them with mining locations. The entire study looks very thoughtful. The description of the methodology is very clear. I think the article still requires a series of corrections, as some elements in the presentation of the results are incomprehensible to the reader (some very obvious elements to the authors are not understandable to the reader). In particular, the authors should correct the explanations of abbreviations, figure captions, and map legends. I suggest the authors think about the number of figures in the article. I believe that some of the figures can be combined into one (details in the attached file). The last part is the strong point of the article. Apart from the classical summary, the authors added recommendations that constitute a practical aspect of the research.

Round 2

Reviewer 2 Report

The authors have greatly improved their article. All my comments have been studied and relevant elements have been added in the introduction and methodology. The figures have also been revised. I think the paper is now ready for publication.